# DEEPENING HIDDEN REPRESENTATIONS FROM PRE-TRAINED LANGUAGE MODELS

## ABSTRACT

Transformer-based pre-trained language models have proven to be effective for learning contextualized language representation. However, current approaches only take advantage of the output of the encoder's final layer when fine-tuning the downstream tasks. We argue that only taking single layer's output restricts the power of pre-trained representation. Thus we deepen the representation learned by the model by fusing the hidden representation in terms of an explicit HIdden Representation Extractor (HIRE), which automatically absorbs the complementary representation with respect to the output from the final layer. Utilizing RoBERTa as the backbone encoder, our proposed improvement over the pre-trained models is shown effective on multiple natural language understanding tasks and help our model rival with the state-of-the-art models on the GLUE benchmark.

## 1 INTRODUCTION

Language representation is essential to the understanding of text. Recently, pre-trained language models based on Transformer (Vaswani et al., 2017) such as GPT (Radford et al., 2018), BERT (Devlin et al., 2019), XLNet (Yang et al., 2019), and RoBERTa (Liu et al., 2019c) have been shown to be effective for learning contextualized language representation. These models have since continued to achieve new state-of-the-art results on a variety of natural language processing tasks. They include question answering (Rajpurkar et al., 2018; Lai et al., 2017), natural language inference (Williams et al., 2018; Bowman et al., 2015), named entity recognition (Tjong Kim Sang & De Meulder, 2003), sentiment analysis (Socher et al., 2013) and semantic textual similarity (Cer et al., 2017; Dolan & Brockett, 2005).

Normally, Transformer-based models are pre-trained on large-scale unlabeled corpus in an unsupervised manner, and then fine-tuned on the downstream tasks through introducing task-specific output layer. When fine-tuning on the supervised downstream tasks, the models pass directly the output of Transformer encoder's final layer, which is considered as the contextualized representation of input text, to the task-specific layer.

However, due to the numerous layers (i.e., Transformer blocks) and considerable depth of these pre-trained models, we argue that the output of the last layer may not always be the best representation of the input text during the fine-tuning for downstream tasks. Devlin et al. (2019) shows diverse combinations of different layers' outputs of the pre-trained BERT result in distinct performance on CoNLL-2003 Named Entity Recognition (NER) task (Tjong Kim Sang & De Meulder, 2003). Peters et al. (2018b) points out for pre-trained language models, including Transformer, the most transferable contextualized representations of input text tend to occur in the middle layers, while the top layers specialize for language modeling. Therefore, the onefold use of the last layer's output may restrict the power of the pre-trained representation.

In this paper, we propose an extra network component design for Transformer-based model, which is capable of adaptively leveraging the hidden information in the Transformer's hidden layers to refine the language representation. Our introduced additional components include two main additional components:

1. HIdden Representation Extractor (HIRE) dynamically learns a complementary representation which contains the information that the final layer's output fails to capture.

2. Fusion network integrates the hidden information extracted by the HIRE with Transformer final layer's output through two steps of functionalities, leading to a refined contextualized language representation.

Taking advantage of the robustness of RoBERTa by using it as our backbone Transformer-based encoder (Liu et al., 2019c), we conduct experiments on GLUE benchmark (Wang et al., 2018), which consists of nine Natural Language Understanding (NLU) tasks. With the help of HIRE, our model outperforms the baseline on 5/9 of them and advances the state-of-the-art on SST-2 dataset. Keeping the backbone Transformer model unchanged on its architecture, pre-training procedure and training objectives, we get comparable performance with other state-of-the-art models on the GLUE leaderboard, which verifies the effectiveness of the proposed HIRE enhancement over Transformer model.

## 2 MODEL

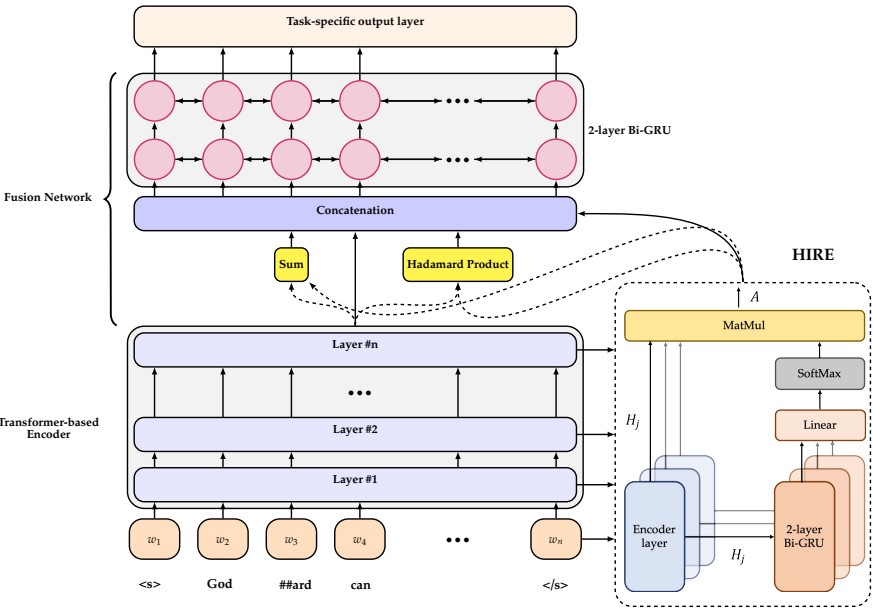

Figure 1: Architecture of our model. HIRE denotes HIdden Representation Extractor, in which the Bi-GRUs share the same parameters for each layer's output.

### 2.1 TRANSFORMER-BASED ENCODER

Transformer-based encoder is responsible for encoding input text into contextualized representation. Let $\{w_1, \ldots, w_n\}$ represent a sequence of $n$ words of input text, Transformer-based encoder encodes the input sequence into its universal contextualized representation $\boldsymbol{R} \in \mathbb{R}^{n \times d}$:

$$\boldsymbol{R} = \text{Encoder}(\{w_1, \ldots, w_n\}) \tag{1}$$

where $d$ is the hidden size of the encoder and $\boldsymbol{R}$ is the output of Transformer-based encoder's last layer which has the same length as the input text. We call it **preliminary representation** in this paper to distinguish it with the one that we introduce in section 2.2. Here, we omit a rather extensive formulations of Transformer and refer readers to Vaswani et al. (2017), Radford et al. (2018) and Devlin et al. (2019) for more details.

### 2.2 HIDDEN REPRESENTATION EXTRACTOR

Transformer-based encoder normally has many structure-identical layers stacked together, for example, BERT$_{\text{LARGE}}$ and XLNet$_{\text{LARGE}}$ all contain 24 layers of the identical structure, either outputs

from these hidden layers or the last layer, but not only limited to the latter, may be extremely helpful for specific downstream task. To make full use of the representations from these hidden layers, we introduce an extra component attached to the original encoder, HIdden Representation Extractor (HIRE) to capture the complementary information that the output of the last layer fails to capture. Since each layer does not take the same importance to represent a certain input sequence for different downstream tasks, we design an adaptive mechanism that can compute the importance dynamically. We measure the importance by an **importance score**.

The input to the HIRE is $\{\boldsymbol{H}_0, \dots, \boldsymbol{H}_j, \dots, \boldsymbol{H}_l\}$ where $l$ represents the number of layers in the encoder. Here $\boldsymbol{H}_0$ is the initial embedding of input text, which is the input of the encoder's first layer but is updated during training and $\boldsymbol{H}_j \in \mathbb{R}^{n \times d}$ is the hidden-state of the encoder at the output of layer $j$. For the sake of simplicity, we call them all hidden-state afterwards.

We use the same 2-layer Bidirectional Gated Recurrent Unit (GRU) (Cho et al., 2014) to summarize each hidden-state of the encoder. Instead of taking the whole output of GRU as the representation of the hidden state, we concatenate GRU's each layer and each direction's final state together. In this way, we manage to summarize the hidden-state into a fixed-sized vector. Hence, we obtain $\boldsymbol{U} \in \mathbb{R}^{(l+1) \times 4d}$ with $\boldsymbol{u}_i$ the summarized vector of $\boldsymbol{H}_i$:

$$\boldsymbol{u}_i = \text{Bi-GRU}(\boldsymbol{H}_i) \in \mathbb{R}^{4d} \tag{2}$$

where $0 \le i \le l$. Then the importance value $\alpha_i$ for hidden-state $\boldsymbol{H}_i$ is calculated by:

$$\alpha_i = \text{ReLU}(\boldsymbol{W}^T \boldsymbol{u}_i + b) \in \mathbb{R} \tag{3}$$

where $\boldsymbol{W} \in \mathbb{R}^{4d \times 1}$ and $b \in \mathbb{R}$ are trainable parameters. Let $\boldsymbol{\alpha}=\{\alpha_i\}$ be normalized into a probability distribution $\boldsymbol{s}$ through a softmax layer:

$$\boldsymbol{s} = \text{softmax}(\boldsymbol{\alpha}) \in \mathbb{R}^{l+1} \tag{4}$$

where $s_i$ is the normalized weight of hidden-state $i$ when computing the representation. Subsequently, we obtain the input sequence's new representation $\boldsymbol{A}$ by:

$$\boldsymbol{A} = \sum_{i=0}^{l+1} s_i \boldsymbol{H}_i \in \mathbb{R}^{n \times d} \tag{5}$$

With the same shape as the output of Transformer-based encoder's final layer, HIRE's output $\boldsymbol{A}$ is expected to contain the additional useful information from the encoder's hidden-states and we call it **complementary representation**.

## 2.3 FUSION NETWORK

This module fuses the information contained in the output of Transformer-based encoder and the one extracted from encoders' hidden states by HIRE.

Given the preliminary representation $\boldsymbol{R}$, instead of letting it flow directly into task-specfic output layer, we combine it together with the complementary representation $\boldsymbol{A}$ to yield $\boldsymbol{M}$, defined by:

$$\boldsymbol{M} = [\boldsymbol{R}; \boldsymbol{A}; \boldsymbol{R} + \boldsymbol{A}; \boldsymbol{R} \circ \boldsymbol{A}] \in \mathbb{R}^{n \times 4d} \tag{6}$$

where $\circ$ is element-wise multiplication (Hadamard Product) and $[; ]$ is concatenation across the last dimension.

Later, two-layer bidirectional GRU, with the output size of $d$ for each direction, is used to fully fuse the information contained in the preliminary representation and the complementary representation. We concatenate the outputs of the GRUs in two dimensions together for the final contextualized representation:

$$\boldsymbol{F} = \text{Bi-GRU}(\boldsymbol{M}) \in \mathbb{R}^{n \times 2d} \tag{7}$$

## 2.4 OUTPUT LAYER

The output layer is task-specific. The following are the concerned implementation details on two tasks, classification and regression.

For classification task, given the input text's contextualized representation $F$, following Devlin et al. (2019), we take the first row $c \in \mathbb{R}^{2d}$ of $F$ corresponding to the first input token ($< s >$) as the aggregated representation. Let $m$ be the number of labels in the datasets, we pass $c$ through a feed-forward network (FFN):

$$q = W_2^T \cdot \tanh(W_1^T c + b_1) + b_2 \in \mathbb{R}^m \tag{8}$$

with $W_1 \in \mathbb{R}^{2d \times d}$, $W_2 \in \mathbb{R}^{d \times m}$, $b_1 \in \mathbb{R}^d$ and $b_2 \in \mathbb{R}^m$ the only parameters that we introduce in output layer. Finally, the probability distribution of predicted label is computed as:

$$p = \text{softmax}(q) \in \mathbb{R}^m \tag{9}$$

For regression task, we obtain $q$ in the same manner with $m = 1$, and take $q$ as the predicted value.

## 2.5 TRAINING

For classification task, the training loss to be minimized is defined by the Cross-Entropy:

$$L(\boldsymbol{\theta}) = -\frac{1}{T} \sum_{i=1}^{T} \log(p_{i,c}) \tag{10}$$

where $\boldsymbol{\theta}$ is the set of all parameters in the model, $T$ is the number of examples in the dataset, $p_{i,c}$ is the predicted probability of gold class $c$ for example $i$ .

For regression task, we define the training loss by mean squared error (MSE):

$$L(\boldsymbol{\theta}) = \frac{1}{T} \sum_{i=1}^{T} (q_i - y_i)^2 \tag{11}$$

where $q_i$ is the predicted value for example $i$ and $y_i$ is the ground truth value for example $i$.

## 3 EXPERIMENTS

### 3.1 DATASET

We conducted the experiments on the General Language Understanding Evaluation (GLUE) benchmark (Wang et al., 2018) to evaluate our proposed method. GLUE is a collection of 9 diverse datasets[1] for training, evaluating, and analyzing natural language understanding models.

### 3.2 MAIN RESULTS

| Model | Single Sentence | | Similarity and Paraphrase | | | Natural Language Inference | | |
|---|---|---|---|---|---|---|---|---|
| | CoLA (Mc) | SST-2 (Acc) | MRPC (Acc) | QQP (Acc) | STS-B (Pearson) | MNLI-m/mm (Acc) | QNLI (Acc) | RTE (Acc) |
| MT-DNN | 63.5 | 94.3 | 87.5 | 91.9 | 90.7 | 87.1/86.7 | 92.9 | 83.4 |
| XLNET-large | 69.0 | 97.0 | 90.8 | 92.3 | 92.5 | 90.8/90.8 | 94.9 | 85.9 |
| ALBERT-xxlarge (1.5M) | 71.4 | 96.9 | 90.9 | 92.2 | 93.0 | 90.8/- | 95.3 | 89.2 |
| RoBERTa-large | 68.0 | 96.4 | 90.9 | 92.2 | 92.4 | 90.2/90.2 | 94.7 | 86.6 |
| **RoBERTa+HIRE** | 69.7 | 96.8 | 90.9 | 92.0 | 92.4 | 90.7/90.4 | 95.0 | 86.6 |
| | *(+1.7)* | *(+0.4)* | - | *(-0.2)* | - | *(+0.5/+0.2)* | *(+0.3)* | - |

Table 1: GLUE Dev results. Our results are based on single model trained with single task and a median over five runs with different random seed but the same hyperparameter is reported for each task. The results of MT-DNN, XLNET$_{\text{LARGE}}$, ALBERT and RoBERTa are from Liu et al. (2019b), Yang et al. (2019), Lan et al. (2020) and Liu et al. (2019c). See the lower-most row for the performance of our approach. Mc, acc and pearson denote Matthews correlation, accuracy and Person correlation coefficient respectively.

---

[1] All the datasets can be obtained from `https://gluebenchmark.com/tasks`

| Model | Params (M) | Shared (M) |
|---|---|---|
| ALBERT-xxlarge | 235 | - |
| MT-DNN | 350 | 340 |
| RoBERTa | 355 | - |
| RoBERTa+HIRE | 437 | - |

Table 2: Parameter Comparison.

Table 1 compares our method with a list of Transformer-based models on the development set. Model parameter comparison is shown in Table 2. To obtain a direct and fair comparison with our baseline model RoBERTa, following the original paper (Liu et al., 2019c), we fine-tune RoBERTa+HIRE separately for each of the GLUE tasks, using only task-specific training data. The single-model results for each task are reported. We run our model with five different random seeds but the same hyperparameters and take the median value. Due to the problematic nature of WNLI dataset, we exclude its results in this table. The results shows that RoBERTa+HIRE consistently outperforms RoBERTa on 4 of the GLUE task development sets, with an improvement of 1.7 points, 0.4 points, 0.5/0.2 points, 0.3 points on CoLA, SST-2, MNLI and QNLI respectively. And on the MRPC, STS-B and RTE task, our model get the same result as RoBERTa. It should be noted that the improvement is entirely attributed to the introduction of HIdden Representation Extractor and fusion network in our model.

| Model | Single Sentence | | Similarity and Paraphrase | | | Natural Language Inference | | | | Score |
|---|---|---|---|---|---|---|---|---|---|---|
| | CoLA | SST-2 | MRPC | QQP | STS-B | MNLI-m/mm | QNLI | RTE | WNLI | |
| | 8.5k | 67k | 3.7k | 364k | 7k | 393k | 108k | 2.5k | 634 | |
| BERT[a] | 60.5 | 94.9 | 89.3/85.4 | 72.1/89.3 | 87.6/86.5 | 86.7/85.9 | 92.7 | 70.1 | 65.1 | 80.5 |
| MT-DNN[b] | 68.4 | 96.5 | 92.7/90.3 | 73.7/89.9 | 91.1/90.7 | 87.9/87.4 | **96.0** | 86.3 | 89.0 | 87.6 |
| FreeLB-RoBERTa[c] | 68.0 | 96.8 | 93.1/90.8 | **74.8**/90.3 | 92.3/92.1 | 91.1/90.7 | 95.6 | 88.7 | 89.0 | **88.4** |
| XLNet[d] | **70.2** | **97.1** | 92.9/90.5 | 74.7/90.4 | **93.0/92.6** | 90.9/90.9 | - | 88.5 | **92.5** | - |
| ALBERT[e] | 69.1 | **97.1** | **93.4/91.2** | 74.2/**90.5** | 92.5/92.0 | **91.3/91.0** | - | **89.2** | 91.8 | - |
| RoBERTa[f] | 67.8 | 96.7 | 92.3/89.8 | 74.3/90.2 | 92.2/91.9 | 90.8/90.2 | 95.4 | 88.2 | 89.0 | 88.1 |
| **RoBERTa+HIRE** | 68.6 | **97.1** | 93.0/90.7 | 74.3/90.2 | 92.4/92.0 | 90.7/90.4 | 95.5 | 87.9 | 89.0 | 88.3 |
| | (+0.8) | (+0.4) | (+0.7/+0.9) | - | (+0.2/+0.1) | (-0.1/+0.2) | (+0.1) | (-0.3) | - | (+0.2) |

Table 3: GLUE Test results, scored by the official evaluation server. All the results are obtained from GLUE leaderboard (`https://gluebenchmark.com/leaderboard`). The number below each task's name indicates the size of training dataset. Recently the GLUE benchmark has forbidden all the submissions to treat the QNLI as a ranking task, which results in the missing of some models' accuracies on the QNLI. [a]Devlin et al. (2019); [b]Liu et al. (2019b); [c]Zhu et al. (2020); [d]Yang et al. (2019); [e]Lan et al. (2020); [f]Liu et al. (2019c).

Table 3 presents the results of HIRE enhancement and other models on the test set that have been submitted to the GLUE leaderboard. Following Liu et al. (2019c), we fine-tune STS-B and MRPC starting from the MNLI single-task model. Given the simplicity between RTE, WNLI and MNLI, and the large-scale nature of MNLI dataset (393k), we also initialize RoBERTa+HIRE with the weights of MNLI single-task model before fine-tuning on RTE and WNLI. We submitted the ensemble-model results to the leaderboard. The results show that RoBERTa+HIRE still boosts the strong RoBERTa baseline model on the test set. To be specific, RoBERTa+HIRE outperforms RoBERTa over CoLA, SST-2, MRPC, SST-B, MNLI-mm and QNLI with an improvement of 0.8 points, 0.4 points, 0.7/0.9 points, 0.2/0.1 points, 0.2 points and 0.1 points respectively. In the meantime, RoBERTa+HIRE gets the same results as RoBERTa on QQP and WNLI. By category, RoBERTa+HIRE has better performance than RoBERTa on the single sentence tasks, similarity and paraphrase tasks. It is worth noting that our model obtains state-of-the-art results on SST-2 dataset, with a score of 97.1. The results are quite promising since HIRE does not modify the encoder internal architecture (Yang et al., 2019) or redefine the pre-training procedure (Liu et al., 2019c) , getting the comparable results with them.

## 4 ABLATION STUDY

In this section, we perform a set of ablation experiments to understand the effects of our proposed techniques during fine-tuning. All the results reported in this section are a median of five random runs.

### 4.1 MODEL DESIGN CONSIDERATION

| Model | CoLA (Mc) | STS-B (Pearson) |
|---|---|---|
| Our model | 69.7 | 92.4 |
| w/o HIRE | 68.5 | 91.5 |
| w/o Fusion network | 68.2 | 92.2 |

Table 4: Ablation study over model design consideration on the development set of CoLA and STS-B. The result for each model is a median of five random runs. Both HIRE and fusion network significantly improve the model performance on all two datasets.

To individually evaluate the importance of HIRE and fusion network, we vary our model in the following ways and conduct the experiments on the development set of CoLA and STS-B. The results are in Table 4:

- We remove the HIRE from our model and pass the preliminary representation $R$ directly into the fusion network. In order to keep the fusion network, we fuse the preliminary representation with itself, which means we define instead $M$ by:

$$M = [R; R; R + R; R \circ R] \in \mathbb{R}^{n \times 4d} \tag{12}$$

  The results are presented in row 2.
- We remove the fusion network, and take the outputs of HIRE as the final representation of the input and flow it directly into the output layer and present the results in row 3.

As can be seen from the table, to use of HIRE to extract the hidden information is crucial to the model: Matthews correlation of CoLA and Pearson correlation coefficient of STS-B drop dramatically by 1.2 and 0.9 points if it's removed. We also observe that fusion network is an important component that contributes 1.5/0.2 gains on the CoLA and STS-B tasks. The results are aligned with our assumption that HIRE extracts the complementary information from the hidden layers while the fusion network fuses it with the preliminary representation.

### 4.2 EFFECT OF DYNAMIC MECHANISM

| Method | Mc | |
|---|---|---|
| HIRE (dynamic) | 69.7 | |
| mean, all 25 layers | 68.9 | *(-0.8)* |
| mean, last 6 layers | 68.0 | *(-1.7)* |
| mean, first 6 layers | 68.8 | *(-0.9)* |
| random, all 25 layers | 69.3 | *(-0.4)* |
| random, last 6 layers | 68.1 | *(-1.6)* |

Table 5: Effect of dynamic mechanism when computing the importance scores. A median Matthews correlation of five random runs is reported for CoLA on the development set.

We investigate how the adaptive assignment mechanism of the importance score affects the model's performance. CoLA is chosen as the downstream task. Table 5 compares our proposed mechanism

| Number | 0 | 1 | 2 | 3 |
|--------|------|------|--------|------|
| Mc | 68.5 | 69.3 | **69.7** | 68.5 |

Table 6: Ablation study over GRU layer number in fusion network on the development set of CoLA. The results are a median Matthews correlation of five random runs. The best result is in **bold**.

with diverse fixed combinations of importance scores over different range of layers: first 6, last 6 or all 25 layers. Two strategies are studied: *mean* and *random*. In the *mean* situation, we suppose all the layers contribute exactly the same. On the contrary, under the *random* condition, we generate a score for each layer randomly and do the SoftMax operation across all the layers.

From the table, we observe that fixing each layer's importance score for all examples hurts the model performance. Across all rows, we find that none of single strategy can yield the best performance. The results enable us to conclude that the hidden-state weighting mechanism introduced by HIRE may indeed adapt to diverse downstream tasks for better performance.

### 4.3 EFFECT OF GRU LAYER NUMBER IN FUSION NETWORK

To investigate the effect of GRU layer number in the fusion network, we set the GRU layer number from 0 to 3 and conduct the ablation experiments on the development dataset of CoLA. Table 6 shows the results. We observe that the modest number 2 would be a better choice.

## 5 ANALYSIS

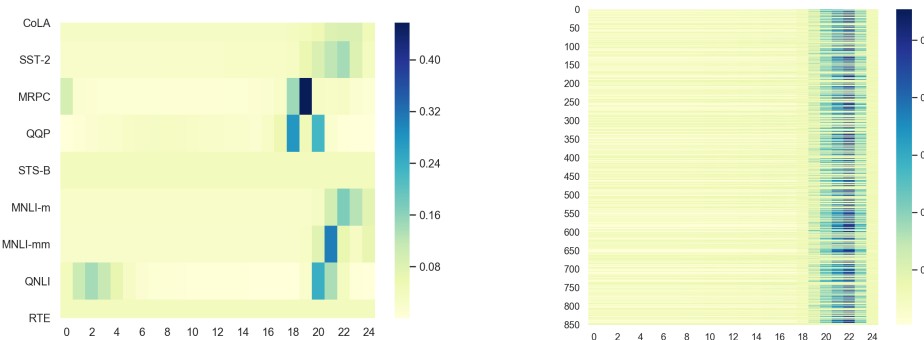

Figure 2: **Left**: Distribution of importance scores over different layers when computing the complementary representation for various NLU tasks. The numbers on the abscissa axis indicate the corresponding layer with 0 being the first layer and 24 being the last layer. **Right**: Distribution of importance scores over different layers for each example of SST-2 dataset. The number on the ordinate axis denotes the index of the example.

We compare the importance score's distribution of different NLU tasks. For each task, we run our best single model over the development set and the results are calculated by averaging the values across all the examples within each dataset. The results are presented in the left of Figure 2. From the top to the bottom of the heatmap, the results are placed in the following order: single-sentence tasks, similarity and paraphrase tasks, and natural language inference tasks. From it, we find that the distribution differs among the different tasks, which demonstrates HIRE's dynamic ability to adapt for distinct tasks when computing the complimentary representation. The most important contribution occurs below the final layer for all the tasks except STS-B and RTE. All layers have a close contribution for STS-B and RTE task. In addition, we find that QNLI, a classification task which requires the model to determine whether the context sentence contains the answer to the question, relies on layers 2-4 to some extent. In fact, Jawahar et al. (2019) state that BERT mostly captures phrase-level (or span-level) information in the lower layers and that this information gets gradually diluted in higher layers. Since QNLI is derived from the span extraction question answering dataset

SQuAD, where the answer is a span of the input text, we speculate that HIRE takes advantage of this information when making the prediction.

The right of Figure 2 presents the distribution of importance scores over different layers for each example of SST-2 dataset. The number on the ordinate axis denotes the index of the example. It shows that HIRE can adapt not only distinct tasks but also the different examples. In the meantime, we observe also that even though there are subtle differences among these examples, they follow certain same patterns when calculating the complementary representation, for example, layers 21 and 22 contribute the most for almost all the examples and also the layers around them. But the figure shows also that for some examples, all layers contribute nearly equally.

## 6 RELATED WORK

Transformer model is empowered by self-attention mechanism and has been applied as an effective model architecture design in quite a lot of pre-trained language models (Vaswani et al., 2017). OpenAI GPT (Radford et al., 2018) is the first model that introduced Transformer architecture into unsupervised pre-training. But instead of unidirectional training like GPT, BERT (Devlin et al., 2019) adopts Masked LM objective when pre-training, which enables the representation to incorporate context from both direction. The next sentence prediction (NSP) objective is also used by BERT to better model the relationship between sentences. Trained with dynamic masking, large mini-batches and a larger byte-level BPE, full-sentences without NSP, RoBERTa (Liu et al., 2019c) improves BERT's performance on the downstream tasks from a better BERT re-implementation training on BooksCorpus(Zhu et al., 2015), CC-News, Openwebtext and Stories. In terms of fine-tuning on downstream tasks, all these powerful Transformer-based models help various NLP tasks continuously achieve new state-of-the-art results. Diverse new methods have been proposed recently for fine-tuning the downstream tasks, including multi-task learning (Liu et al., 2019b), adversarial training (Zhu et al., 2020) or incorporating semantic information into language representation (Zhang et al., 2020).

Traditionally, downstream tasks or back-end part of models take representations from the last layer of the pre-trained language models as the default input. However, recent studies trigger researchers' interest on the intermediate layers' representation learned by pre-trained models. When studying the linguistic knowledge and transferability of contextualized word representations with a series of seventeen diverse probing tasks, Liu et al. (2019a) observe that Transformers tend to encode transferable features in their intermediate layers, aligned with the results by Peters et al. (2018b). Jawahar et al. (2019) show a rich hierarchy of linguistic information is encoded by BERT, with surface features, syntactic features and semantic features lying from bottom to top in the layers. So far, to our best knowledge, exploiting the representations learned by hidden layers is still stuck with limited empirical observations, which motivates us to propose a general solution for fully exploiting all levels of representations learned by the pre-trained language models. However, we want to note that a freezing/fine-tuning discrepancy may exist, since previous observations are made by adopting a feature extracting approach while our work is conducted in the fine-tuning scenario.

Several works have made attempts to utilize the information from the intermediate layers of deep models (Zhao et al., 2015; Peters et al., 2018a; Zhu et al., 2018; Tenney et al., 2019). Peters et al. (2018a) propose a deep contextualized word representation called ELMo, which is a task-specific linear combination of the intermediate layer representations in the bidirectional LSTM language model. Tenney et al. (2019) find that using ELMo-style scalar mixing of layer activations, both deep Transformer models (BERT and GPT) gain a significant performance improvement on a novel edge probing task. Zhu et al. (2018) adopt a linear combination of embeddings from different layers in BERT to encode tokens for conversational question answering but in a feature-based way. Compared with these works, our approach still presents the following innovations: 1) Our dynamic layer weighting mechanism calculates importance scores for each example over different layers, which is distinct from previous works (e.g., ELMo), where the contribution score (in the form of a parameter) for each layer is always same for all the examples of the same task. 2) Our approach fuses the representation weighted from the hidden layers and the one from the last layer to get a refined representation. 3) To our best knowledge, our work is the first one that takes advantage of the pre-trained language model's intermediate information in the context of fine-tuning.

## 7 CONCLUSION

This paper presents HIdden Representation Extractor (HIRE), a novel enhancement component that refines language representation by adaptively leveraging the Transformer-based model's hidden layers. In our proposed model design, HIRE dynamically generates complementary representation from all hidden layers other than that from the default last layer. A lite fusion network then incorporates the outputs of HIRE into those of the original model. The experimental results demonstrate the effectiveness of refined language representation for natural language understanding. The analysis highlights the distinct contribution of each layer's output for diverse tasks and different examples.

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

## A  APPENDIX

### A.1  DETAILED DESCRIPTIONS FOR THE GLUE BENCHMARK

Three different tasks are presented in GLUE benchmark (Wang et al., 2018) according to the original paper:

**Single-sentence tasks:** The Corpus of Linguistic Acceptability (**CoLA**) (Warstadt et al., 2018) requires the model to determine whether a sentence is grammatically acceptable; the Stanford Sentiment Treebank (**SST-2**) (Socher et al., 2013) is to predict the sentiment of movie reviews with label of positive or negative.

**Similarity and paraphrase tasks:** Similarity and paraphrase tasks are to predict whether each pair of sentences captures a paraphrase/semantic equivalence relationship. The Microsoft Research Paraphrase Corpus (**MRPC**) (Dolan & Brockett, 2005), the Quora Question Pairs (**QQP**) [2] and the Semantic Textual Similarity Benchmark (**STS-B**) (Cer et al., 2017) are presented in this category.

**Natural Language Inference (NLI) tasks:** Natural language inference is the task of determining whether a "hypothesis" is true (entailment), false (contradiction), or undetermined (neutral) given a "premise". GLUE benchmark contains the following tasks: the Multi-Genre Natural Language Inference Corpus (**MNLI**) (Williams et al., 2018), the converted version of the Stanford Question Answering Dataset (**QNLI**) (Rajpurkar et al., 2016), the Recognizing Textual Entailment (**RTE**) (Dagan et al., 2006; Roy et al., 2006; Giampiccolo et al., 2007; Bentivogli et al., 2009) and the Winograd Schema Challenge (**WNLI**) (Levesque et al., 2012).

Four official metrics are adopted to evaluate the model performance: Matthews correlation (Matthews, 1975), accuracy, F1 score, Pearson and Spearman correlation coefficients.

## A.2 Implementation Details

Our implementation of HIRE and its related fusion network is based on the PyTorch implementation of Transformer [3].

**Preprocessing:** Following Liu et al. (2019c), we adopt GPT-2 (Radford et al., 2019) tokenizer with a Byte-Pair Encoding (BPE) vocabulary of subword units size 50K. The maximum length of input sequence is 128 tokens.

**Model configurations:** We use RoBERTa$_{LARGE}$ as the Transformer-based encoder and load the pre-training weights of RoBERTa (Liu et al., 2019c). Like BERT$_{LARGE}$, RoBERTa$_{LARGE}$ model contains 24 Transformer-blocks, with the hidden size being 1024 and the number of self-attention heads being 16 (Liu et al., 2019c; Devlin et al., 2019).

**Optimization:** We use Adam optimizer (Kingma & Ba, 2015) with $\beta_1 = 0.9$, $\beta_2 = 0.98$ and $\epsilon = 10^{-6}$ and the learning rate is selected amongst {1e-5, 2e-5} with a warmup rate of 0.06 depending on the nature of the task. The number of training epochs ranges from 3 to 27 with the early stop and the batch size is selected amongst {16, 32, 48}. In addition to that, we clip the gradient norm within 1 to prevent exploding gradients problem occurring in the recurrent neural networks in our model.

**Regularization:** We employ two types of regularization methods during training. We apply dropout (Srivastava et al., 2014) of rate 0.1 to all layers in the Transformer-based encoder and GRUs in the HIRE and fusion network. We additionally adopt L2 weight decay of 0.1 during training.

---

[2] https://www.quora.com/q/quoradata/First-Quora-Dataset-Release-Question-Pairs
[3] https://github.com/huggingface/transformers

