# OpenReview forum: "Deepening Hidden Representations from Pre-trained Language Models"
_ICLR.cc/2021/Conference — Reject_

### Official Review · AnonReviewer4 · 2020-10-27
**Initial Review**

**Rating:** 4
**Confidence:** 5

**Review:**


### Summary
In this paper, the authors propose a fine-tuning scheme for pre-trained language models, called Hidden Representation Extractor (HIRE). The proposed method consists of two components.
1) Hidden representation extractor (HIRE): a two-layer bi-GRU model summarizes the hidden representation of the encoder with a weighted-sum scheme.
2) Fusion network: combine the weighed representation with original representation via the bi-GRU model.
The authors apply the proposed approach on top of RoBERTa and evaluated on general benchmark GLUE. The experimental results show that the proposed method could give moderate improvements over RoBERTa baseline system.

The authors' main contribution lies in two components: HIRE and fusion network. However, unfortunately, both components are not novel to me.
1) The idea of HIRE model is very similar to utilizing ELMo in down-stream tasks, where the hidden representations of each layer are weighted.
2) The central part of the fusion network utilizes a concatenation scheme of [A;B;A+B;A*B], which is also a widely used approach, similar to Chen et al. (2017), Seo et al. (2017), Hu et al. (2019), etc.

Besides the novelty of the proposed components, the experiments are also not well-designed. I assume the proposed method could apply to various pre-trained language models (PLMs), as this is an approach that aims to apply on the fine-tuning stage. However, the authors only present the results on top of RoBERTa model, which cannot demonstrate that the proposed model can be generally applied to various PLMs. I would suggest the authors evaluate their approach in a more general way, such as build on top of various PLMs, or apply on PLM in another language. These would have helped to better demonstrate the effectiveness of the proposed method.

Considering the novelty and generalizability of the proposed method, I recommend rejection for this paper.

[1] Chen et al. Enhanced LSTM for Natural Language Inference.
[2] Seo et al. Bidirectional attention flow for machine comprehension.
[3] Hu et al. Read+ verify: Machine reading comprehension with unanswerable questions.


### PROs
1. A possible useful framework for fine-tuning down-stream tasks for pre-trained language models.


### CONs
1. The design of the components is not novel.
2. The experiments are not well-designed, and the results only show marginal improvements over RoBERTa.
3. The generality of the proposed method is not well-studied, as the experiments are only performed on top of RoBERTa.


### Questions
1. Unlike Transformer-based models, RNN models (such as Bi-GRU) are not computationally efficient. How about your training time compared to the baselines? According to Table 2, using HIRE will increase the total parameter from 355 to 437, which will slow down the inference speed.
2. The authors did not cite ELECTRA (Clark et al., 2020) for comparison. After comparing the results, the proposed model is not competitive against ELECTRA, while it would cost more training and inference time, which will weaken the impact of the paper. I strongly recommend the authors also carry out experiments on top of ELECTRA (or ALBERT-xxlarge, etc.) to see if your approach generalizes well on various PLMs.


### Minor Reviews
I did not go through every detail of the writing and only list a few of the issues here.
1. page 1, section 1: CoNNL-2003 -> CoNLL-2003
2. page 1, section 1: (Peter et al., 2018b) points out ... -> Peter et al. (2018b) points out ...
3. page 3, section 2.3: Tansformer -> Transformer; yeild -> yield; elementwise -> element-wise;

---

> ### Author Response · Authors · 2020-11-19
> **Response to Reviewer #4**
>
> Dear Reviewer #4,
>
> Thanks a lot  for going through the paper carefully and providing constructive and useful feedback on our work!
>
> **About the novelty of the proposed components (CON 1)**: Yes, the idea of taking advantage of hidden layers has indeed been seen in several previous works.  But compared with these works, our work still presents the following innovations: 1) Our approach fuses the representation weighted from the hidden layers and the one from the last layer to get a refined representation. 2) Our dynamic layer weighting mechanism calculates importance scores for different examples over different layers (see Figure 3), which is distinct from previous works (e.g., ELMO), where the contribution score (in the form of a parameter) for each layer is always same for all the examples.
>
> **About the significance of the results(CON 2)**:  Our chosen RoBERTa baseline has been very strong. As reviewer #1 points out,
> > RoBERTa already achieves super-human results on the GLUE benchmark, meaning that it is probably already close to some upper limit. This makes it even more difficult for your model to substantially improve upon it.
>
> So we don’t consider that our gain as a small improvement.
>
> **About the generality of the proposed method (CON 3 & Question 2)**:  Due to the limitation of the computation capacity, we have only conducted experiments on the RoBERTa. But we would like to explore the generality of HIRE on ELECTRA if time permits.  Thanks for mentioning the proposition. Since ELECTRA is also a Transformer-based pre-trained language model, we believe that HIRE can further improve its performance.
>
> **About the computation efficiency of RNN (Question 1)**: Yes, the main contribution of parameters is due to GRUs in both HIRE and Fusion-network. The choice of GRU is due to its robustness to encode sequence and the effectiveness to fuse information (Seo et al., 2017, Clark et al., 2017).  But unfortunately, due to the usage of GRUs in our model, the computational time of RoBERTa-HIRE has increased by about 70%. We will explore a light HIRE (by considering other network architecture) later.
>
> **About the typo**: Thank you so much for pointing out the typo for us,  the mistakes have been corrected in the new version.
>
> [1] Seo et al. Bidirectional attention flow for machine comprehension.
>
> [2] Clark et al. Simple and Effective Multi-Paragraph Reading Comprehension

---

### Official Review · AnonReviewer2 · 2020-10-28
**A hidden representation extractor for improving pre-trained representation**

**Rating:** 5
**Confidence:** 4

**Review:**

This paper presents a new mechanism, called HIRE, to extract more information from the intermediate layers of pre-trained models, which will be further fused with the last layer of pre-trained models. The main contribution of this work is the newly proposed dynamic feature extractor HIRE and the fusion network. Experiments confirmed the effectiveness of the proposed method, and some interesting observations on the importance of different layers for different tasks were given (i.e. Figure 2).

However, the proposed HIRE and Fusion modules can be viewed as the combination of some widely used deep learning mechanisms (e.g. Bi-GRU, softmax), using Bi-GRE to represent a sequence is a widely adopted choice, so the contribution in modeling is quite limited. Some existing works, like Self-Adaptive Hierarchical Sentence Model https://arxiv.org/pdf/1504.05070.pdf, have already proposed similar ideas to use more information from the intermediate layers of deep models.

The paper is well-written and organized. I have several concerns:
1. All the models are evaluated on the GLUE dataset, experiments on more challenging tasks like QA (e.g. SQuAD 1.1/2.0) should be added. It would be interesting to visualize the importance scores of each layer for this challenging task.

2. As illustrated in Figure 2, different downstream task highly depends on different intermediate layers, the authors are suggested to conduct experiments under the multi-task setting, since the HIRE can adaptively select the intermediate layers, the proposed method should have a big advantage for this setting.

---

> ### Author Response · Authors · 2020-11-19
> **Response to Reviewer #2**
>
> Dear Reviewer #2,
>
> Thanks a lot for your careful review. Please see our answers below:
>
> **About the ideas to use more information from the intermediate layers of deep models**:  Indeed,  some existing works have already proposed similar ideas to use more information from the intermediate layers of deep models. But different from previous works, we further fuse this information with the one of the last layer. And to our best knowledge, our work is the first one that takes advantage of the intermediate layers in the context of  pre-trained language model fine-tuning.  We have cited the paper you mentioned and added a discussion in Section 6.
>
> **About the experimental results on QA (Concern 1)**: QNLI is the modified version of the QA task SQuAD,  which requires the model to determine whether the context sentence contains the answer to the question. The results show  that we can further improve the performance of RoBERTa on QNLI with HIRE. But we will update our experimental results on the SQuAD dataset and explore the behaviour of importance scores on the original QA task, if time permits.
>
> **About the experiments under the multi-task setting (Concern 2)**: Thanks a lot for your insightful suggestion! Indeed, since our method can adapt to each task dynamically, it could have the potential to have a big advantage under the multi-task setting. We will leave it for future work and we welcome also the community to integrate our idea with the multi-task training.

---

> > ### Comment · AnonReviewer2 · 2020-11-24
> > **Some of my concerns have been addressed**
> >
> > Dear authors,
> >
> > Thanks for your reply. Your explanation well addressed some of my concerns (i.e. concern2), so I have updated my score to 5, while I still do not think the current version has reached the standard to get published in ICLR.

---

### Official Review · AnonReviewer1 · 2020-10-28
**Interesting model, but the experiments are not convincing.**

**Rating:** 6
**Confidence:** 4

**Review:**

The paper proposes a method to improve the downstream performance of a pretrained Transformer on NLP tasks. The core idea is to not only use the output of the last Transformer layer for prediction, but let the model decide how to fuse the information from intermediate layers as well. To dynamically decide which intermediate layers to use depending on the input example, the model uses a mechanism conceptually similar to self-attention, which yields a normalized importance score for each layer. The importance-weighted sum then yields a complementary representation to the last layer. Lastly, another network produces a final, integrated representation from the output at the last layer and the complementary representation, which is then used for prediction.
The model is evaluated on the GLUE benchmark.

Strengths:
The paper is well written and the experimental evaluation seems correct. The paper has a nice ablation study which shows that the learned importance scores, the complementary representations, and the fusion network are needed to reach the model's full performance.

Weaknesses:
The main weakness is that the proposed extension to the baseline is relatively complex and rather heavy-weight in terms of new parameters (introducing ~25% more parameters compared to the baseline according to Table 2), yet only achieves a very marginal relative improvement of 0.2 percent over the baseline. It seems likely that this improvement could be achieved through much simpler means, e.g., additional self-attention layers on top of the last pretrained layer. This is supported by the fact that the paper's analysis of the proposed importance score mechanism doesn't show comprehensible patterns (or the paper doesn't talk about it).

I lean slightly towards rejection, because, although the proposed model is reasonable and could have potential, the experiments do not currently demonstrate that potential, and I hence expect it hard for the community to learn from this paper.

Questions:

* You cite several papers which demonstrate that different layers of pretrained Transformers encode different information, which is the motivation for your architecture. However, the cited sources use a feature extracting approach, i.e., they don't fine tune the encoder on their target task. In the finetuning scenario (yours), can't we expect the model to learn to simply forward the relevant information from intermediate layers to the last layer?

* In Section 5 you analyze which intermediate layers are used for which task. You state that we can see that the model learns to prefer different layers for each task, but don't go into detail. Can you relate the nature of the task to the layers it produces? For example, why does QNLI rely on layers 2-4 to some extent despite being a rather high-level understanding task?

* In Figure 2, right side you observe that different examples tend to different layers. Did you qualitatively inspect the examples and try to find a pattern/link to their preferred intermediate layers?

Suggestions:

* RoBERTa already achieves super-human results on the GLUE benchmark, meaning that it is probably already close to some upper limit. This makes it even more difficult for your model to substantially improve upon it. I would consider evaluating it on the SuperGLUE benchmark instead, since there RoBERTa is not so close to human performance yet.

* Your model is designed to make use of representations at lower layers if a task requires it. But all the tasks in your experiments are evaluated on high-level natural language understanding tasks, which typically require representations at higher layers. This makes it more likely that your model does not improve much over the baseline, because the last layer will arguably already contain much of the information needed for the tasks. I think you should consider different tasks in your experiments to increase the chance of significant improvements.

* In the current state, it is unclear whether your improvements are coming from the specific model you chose or merely from the fact that you added a lot of parameters. As a control, you could include an experiment where you simply add several self-attention layers on top of RoBERTa such that you match the number of parameters of your model.

---

> ### Author Response · Authors · 2020-11-19
> **Response to Reviewer #1**
>
> Dear Reviewer #1,
>
> First of all, thanks a lot for going through the paper so carefully and providing all those constructive comments. Please see our response below:
>
> **About the motivation of HIRE (Question 1)**: Indeed, the papers we cited don’t fine-tune the encoder on their target task to make the observation that different hidden layers have different advantages. Right because of such previous findings, we are motivated to conduct this work to put all these advantages together by our proposed method.  In fact, Figure 2 in section 5 shows exactly that during fine-tuning the model doesn't simply forward the relevant information from intermediate layers to the last layer. Otherwise, the important score would have a dominance on the last layer instead of the intermediate layers.
>
> **About the relation of the nature of the task and the preference of layers (Question 2)**: Yes, it’s interesting that QNLI relies on layers 2-4 to some extent. In fact,  Jawahar et al. (2019) state that BERT mostly captures phrase-level (or span-level) information in the lower layers and that this information gets gradually diluted in higher layers. QNLI is a classification task which requires the model to determine whether the context sentence contains the answer to the question. Since QNLI is derived from the span extraction QA dataset SQuAD, where the answer is a span of the input text, we speculate that the HIRE takes advantage of this information to some extent when making the prediction. The corresponding analysis has been added to Section 5 of the revised version of the paper..
>
> **About the pattern to preferred intermediate layers(Question 3)**: Unfortunately, we haven’t done such qualitative inspection until now. We will do it later. Thanks for pointing it out.
>
> **About suggestion 1**:  Yes, our chosen RoBERTa baseline has been very strong and as you said, already achieves super-human results on the GLUE benchmark when we began our work. In consideration of this fact,  the gain that our model obtained on top of RoBERTa is not small. But certainly, we will consider evaluating our method on the SuperGLUE benchmark if time permits.
>
> **About suggestion 2**:  Thanks for this insightful suggestion and observation!  Indeed, the tasks presented in GLUE are high-level NLU tasks, that’s why in Figure 6, the intermediate layers that HIRE selects reside still on the top area.  We will consider conduct experiments on different level tasks.
>
> **About suggestion 3**:  Yes, the current version of HIRE introduces 25% more parameters.  We will conduct the controlled experiment by following your suggestion with the model of the same number of RoBERTa+HIRE by adding several self-attention layers on top of RoBERTa.  At the same time, We will explore a light HIRE (by using the mechanism which doesn’t introduce additional parameters or fewer parameters) later. However, we do believe that *simple* introducing additional parameters without additional pre-training data doesn’t help, since the number of parameters of the current pre-trained language model is already far greater than the number of data points of the downstream tasks.

---

### Decision · Program_Chairs · 2021-01-07
**Final Decision**

**Decision:**

Reject

**Comment:**

This paper proposes  a new mechanism, called HIRE, to  improve the down-stream performance of a pre-trained Transformer on NLP tasks. Different from directly using the last layer of transformer, the proposed model allows the system to dynamically decide which intermediate layers to use based on the input through some sort of gating. The model is evaluated on GLUE, a benchmark for natural language understanding.  My major concerns are the following
1.  the gating mechanism on using intermediate sentence representation is not new,  as pointed by some reviewers, although its implementation on transformers is still interesting.
2.  the empirical part is not convincing enough:  a) GLUE data set is relatively simple, the authors should try something more complex, b）the improvement over baseline is rather modest, which could be achieved with simpler modification.

I'd suggest to reject this paper.